# Relationship between Non-Invasive Brain Stimulation and Autonomic Nervous System

**DOI:** 10.3390/biomedicines12050972

**Published:** 2024-04-28

**Authors:** Giovanni Messina, Antonietta Monda, Antonietta Messina, Girolamo Di Maio, Vincenzo Monda, Pierpaolo Limone, Anna Dipace, Marcellino Monda, Rita Polito, Fiorenzo Moscatelli

**Affiliations:** 1Department of Experimental Medicine, Section of Human Physiology and Unit of Dietetics and Sports Medicine, Università degli Studi della Campania “Luigi Vanvitelli”, 80131 Naples, Italy; giovanni.messina@unicampania.it (G.M.); antonietta.messina@unicampania.it (A.M.); girolamo.dimaio@unicampania.it (G.D.M.); marcellino.monda@unicampania.it (M.M.); 2Department of Human Sciences and Quality of Life Promotion, San Raffaele Telematic University, 00166 Rome, Italy; antonietta.monda@uniroma5.it; 3Department of Movement Sciences and Wellbeing, University of Naples “Parthenope”, 80133 Naples, Italy; vincenzo.monda@uniparthenope.it; 4Department of Psychology and Education, Pegaso Telematic University, 80143 Naples, Italy; pierpaolo.limone@unipegaso.it (P.L.); anna.dipace@unipegaso.it (A.D.); 5Department of Clinical and Experimental Medicine, University of Foggia, 71122 Foggia, Italy; 6Department of Wellbeing, Nutrition and Sport, Pegaso Telematic University, 80143 Naples, Italy; fiorenzo.moscatelli@unipegaso.it

**Keywords:** TMS, non-invasive brain stimulation, heart rate variability, HRV, ANS, rTMS, tDCS

## Abstract

Non-invasive brain stimulation (NIBS) approaches have seen a rise in utilization in both clinical and basic neuroscience in recent years. Here, we concentrate on the two methods that have received the greatest research: transcranial direct current stimulation (tDCS) and repetitive transcranial magnetic stimulation (rTMS). Both approaches have yielded pertinent data regarding the cortical excitability in subjects in good health as well as pertinent advancements in the management of various clinical disorders. NIBS is a helpful method for comprehending the cortical control of the ANS. Previous research has shown that there are notable changes in muscular sympathetic nerve activity when the motor cortex is modulated. Furthermore, in NIBS investigations, the ANS has been employed more frequently as an outcome measure to comprehend the overall impacts of these methods, including their safety profile. Though there is ample proof that brain stimulation has autonomic effects on animals, new research on the connection between NIBS and the ANS has produced contradictory findings. In order to better understand NIBS processes and ANS function, it is crucial to take into account the reciprocal relationship that exists between central modulation and ANS function.

## 1. Introduction

Noninvasive brain stimulation (NIBS) methods have grown in significance in cognitive neuroscience during the last thirty years. By establishing the causal relationships between cortical brain regions and their corresponding cognitive, emotional, sensory, and motor processes, researchers made important advancements [1]. In this review we assess the effect of NIBS on various markers of autonomic nervous system activity, such as heart rate (HR) and various heart rate variability (HRV) parameters [2].

To design future studies that aim to examine the fundamental processes of the cortical modulation of autonomic nervous system (ANS) activities, it is imperative to comprehend whether and how NIBS should be employed to modulate HRV and HR. There are interactions between multiple cortical regions, such as the insular cortex and the medial prefrontal cortex (MPFC), higher subcortical regions, such as the amygdala and the bed nucleus of the stria terminalis (BNST), multiple areas and nuclei of the hypothalamus (e.g., the paraventricular nucleus and dorsomedial hypothalamic nucleus), the periaqueductal gray (PAG), and brainstem regions, such as the parabrachial nucleus (PBN), the solitarius tract (NTS), the nucleus ambiguous (NA), the area postrema, the locus coeruleus, and the dorsal motor nucleus of the vagus (DMNV), and the rostral (RVLM) and caudal (CVLM) ventrolateral medulla. These interconnected regions come together to form the central autonomic nervous system (CAN) [3,4]. Thus, by integrating both cortical perceptual representations of one’s bodily or visceral states and conceptual interpretations of this perceptual input, the MPFC, including the anterior cingulate cortex (ACC) and the insular cortex, is involved in the regulation of hierarchically lower level regions of the CAN such as the amygdala [5]. The entire cortex and subcortical regions share strong connections with the amygdala, which has been shown to be vitally engaged in identifying novel and important stimuli [6]. Efferent projections to the hypothalamic nuclei and the PAG initiate autonomic and behavioral responses to the perceived stimuli, while afferent outputs regulate the attention to and the cognitive processing of those stimuli [7,8]. Bidirectional connections allow the hypothalamus and PAG to transmit signals from the amygdala to the medulla and lower brainstem nuclei [9]. Apart from projections originating from the PAG and hypothalamus, the brainstem also receives visceral afferents that travel via cranial, sacral, and thoracolumbar neural pathways to reach the NTS. The insula, the amygdala, and the hypothalamus receive the signals after they are sent to the PBN [1].

Some of the brain activity that was previously thought to be exclusively related to cognitive functions may be related to autonomic processing. Autonomic processing, via cortico-subcortical pathways, generates physiological responses for behavior that are contextually adaptive and suitable for these higher-order functions [10,11]. An excellent method for explaining the connection between HRV and cognitive functions is the neurovisceral integration model, which postulates that inhibitory pathways in the pre-frontal cortex regulate subcortical cardioaccelerator circuits of the CAN in response to activity in various pre-frontal cortex regions during emotion, self-regulation, and stress.

Prior research has demonstrated that, through network-level effects, the effects of NIBS extend beyond the cortical target regions to subcortical regions such the striatum or thalamus [12,13]. NIBS offers a promising chance to methodically investigate the underlying mechanisms of the correlations between neuronal activity and both HR and HRV found in early brain imaging studies, since it can modulate the neuronal activity of both cortico-subcortical networks and circumscribed cortical regions [1].

Recent studies on the neural regulation of the autonomic nervous system (ANS) also imply that a variety of brain regions act as moderators of ANS activity and, therefore, of heart rate (HR) and heart rate variability (HRV), and that brain regions not traditionally defined as part of the CAN (e.g., the motor cortex, hippocampus, precuneus, lingual gyrus, etc.) are also involved in its complex interactions [14,15]. The degree to which a certain brain region—or even a subregion within that brain region—contributes to ANS activity may vary depending on the problem to be addressed or the environment to be adapted to, according to the findings of two recent meta-analyses [16,17]. Other authors provided an instructive study in this regard, using fMRI to identify brain regions connected to HRV that are independent from task performance, as well as areas associated with changes in HRV during specific task execution [18]. The authors found that changes in HRV were correlated with activity in the orbitofrontal areas during an n-back task and the somatomotor areas during an isometric handgrip exercise. Moreover, Sclocco et al. found, for example, that parasympathetic outflow during motion sickness is moderated by activity in the visual cortex, as indicated by the high-frequency range of HRV [19]. These findings have led to the hypothesis that some of the brain activity previously thought to be exclusively associated with cognitive functions actually relates to autonomic processing. Autonomic processing, via cortico-subcortical pathways, generates physiological responses for behavior that is contextually appropriate for these higher-order functions [15,16,17,18]. An excellent method for explaining the connection between HRV and cognitive functions is the neurovisceral integration model, which postulates that subcortical cardioacceleratory circuits of the CAN are modulated by inhibitory pathways resulting from activity in distinct prefrontal cortex areas during emotion and self-regulation [19,20,21]. However, these human assumptions and models are nearly entirely derived from correlational data obtained from brain imaging or lesion research. Prior research has demonstrated that, through network-level effects, the effects of NIBS extend beyond the cortical target regions to subcortical regions such the striatum or thalamus [22]. NIBS offers a promising chance to methodically look into the underlying mechanisms of the correlations between neuronal activity and both HR and HRV found in early brain imaging studies, since it can modulate the neuronal activity of both cortico-subcortical networks and circumscribed cortical regions. In recent years, there has been a rise in publications due to the promise of NIBS. The rise in empirical research offers the chance to assess whether the ANS is regulated by cortical mechanisms and whether NIBS is a useful technique for altering HR and HRV through meta-analysis.

In this review, we will try to explain the relationships between HRV and NIBS and how intracranial stimulations can interact and, therefore, modulate the ANS. Therefore, this review summarizes the understanding of the research and clinical uses between non-invasive brain stimulation and HRV by searching related keywords in PubMed, Elsevier, and the MDPI database, except for some major references. The search was carried out by matching the following keywords: TMS, HRV, Transcranial magnetic stimulation, Stress.

## 2. Stress and HRV

“A response to change in order to maintain the state of stability or homology that the body has maintained against the stimulus to break the mental and physical balance and stability of the body” is how Hans Selye defined stress [23]. Kenneth Hambly further described stress as a maladaptive state characterized by an overactive sympathetic nervous system that impairs behavior, psychology, and physical health either acutely or over time [24]. Because of several barriers, finding stress biomarkers is still a difficult challenge for academics and doctors. One challenge is the absence of agreement on what constitutes stress. Furthermore, there is currently no thorough framework available for studying how organisms adapt to and operate in continually changing settings [25]. As of right now, there is no accepted benchmark for evaluating stress. Numerous studies have looked at biological markers like cortisol and amylase as well as existing stress-measuring techniques like psychological measures of stress. Furthermore, research on stress and HRV is becoming more common. HRV is defined as the variation in heartbeat interval length [26]. The heart’s response to a range of physiological and environmental events is reflected in its heart rate variability (HRV) [27]. A low HRV indicates a heartbeat that is monotonously regular. Moreover, low HRV is linked to compromised autonomic nerve system (ANS) homeostatic and regulatory processes, which lessen the body’s capacity to withstand both internal and external stresses. As a result, HRV is a noninvasive electrocardiographic technique that may be applied in a range of therapeutic contexts (such as psychological stress assessments) to assess the autonomic nervous system [6]. Under the presumption that HRV is a trustworthy indicator of stress, numerous researchers have carried out studies using HRV to evaluate stress. Few studies, meanwhile, have verified if HRV is a reliable measure of stress. In this review, we looked at the research supporting the use of HRV as a trustworthy measure of psychological stress. To facilitate its future clinical usage as a straightforward, noninvasive diagnostic tool, the usefulness of HRV as a stress indicator needs to be evaluated (Table 1 and Table 2).

The hypothalamic–pituitary–adrenal (HPA) axis and the sympathetic nervous system (SNS) are the two primary mechanisms via which psychological stress affects the body [28]. There is strong coordination and connectivity between the ANS and the HPA axis [29]. Via the sympathetic nervous system (SNS) and the parasympathetic nervous system (PNS), the ANS rapidly induces physiological changes. By removing the inhibitory effect, the PNS encourages the sympathetic reaction to stress, also known as the “fight or flight” response [30]. The release of noradrenaline from the locus coeruleus is one of the subsequent alterations [31,32]. The HPA axis initiates a cascade of endocrine modifications during the stress response, commencing with the hypothalamus’ production of corticotropin-releasing hormone [33]. By blocking the SNS and HPA axis, the PNS contributes significantly to reducing an individual’s stress response [8]. Autonomic activity fluctuations linked to stress interfere with homeostatic functions [30]. The demands of both external stimuli and internal viscera are met by the ANS. While the stress response gives priority to external stimuli above internal needs, homeostasis is linked to the regulation of interior viscera. Therefore, stress happens when the PNS is unable to sufficiently meet an organism’s physiological needs. As a result, parasympathetic tone measurement could be used as a stress and stress vulnerability index. Furthermore, stasis—a deficiency in endogenous variability in peripheral neurally controlled systems, such as heart rate—indicates a serious state of physiological distress [30].

The baroreceptor-heart rate reflex, a reciprocal alteration in ANS activity, serves as an example of the homeostatic functions of the ANS [28,34,35,36,37,38]. At the brainstem level, baroreceptor reflexes are primarily arranged within lower central autonomic substrates. However, these lower autonomic systems were merged with higher neural networks as a result of the rostral brain systems’ evolutionary growth [32]. It has been demonstrated that the limbic system and forebrain, which include the hypothalamus, amygdala, and medial prefrontal cortex, send synaptic projections to both autonomic areas and brainstem reflex networks [31]. Baroreceptor reflex set points can be shifted or inhibited as a result of stressors [34]. The SNS and PNS are the two autonomic branches that are reciprocally controlled by brainstem baroreceptor reflexes [32]. Depending on the situation, the reciprocal method of autonomic control may change. Kim et al. [33] produced different patterns of control with orthostatic challenges and normal psychological stressors using selective pharmacological blockades targeting the SNS and the PNS.

Individual differences in responses to orthostatic stress were negligible. On the other hand, autonomic responsiveness varied widely among individuals in reaction to psychological stressors. Participants exhibited a reciprocal pattern of autonomic response, sympathetic activation, or vagal withdrawal in response to psychological stressors [33]. These results demonstrate that people react differently to psychological pressures.

## 3. Psychological Stress and Autonomic Regulation

There are copious amounts of comprehensive evidence supporting the involvement of the ANS in stress and health. The two main branches of the autonomic nervous system (ANS) are the parasympathetic nervous system (PNS), which is linked to rest, digest, vegetative, and restorative processes, and the sympathetic nervous system (SNS), which is involved with energy mobilization and the fight-or-flight response. These systems are often in a state of dynamic equilibrium when health is good, with the PNS predominating. But, as previously mentioned, an imbalance resulting in the continuous activation of fight-or-flight reactions under stressful situations can cause excessive strain on physiological systems (allostatic load) [30]. The baroreflex is one mechanism that connects the ANS and blood pressure [32]. To maintain blood flow to essential organs like the brain and heart, pressure-sensitive receptors in the carotid and aortic arches sense increases and decreases in blood pressure. These receptors then transmit those signals to the brain, which causes reflex adjustments in blood pressure through the regulation of sympathetic and parasympathetic outflows. Thus, the baroreflex is how ANS activity, as measured by myocardial contractility, peripheral vascular resistance, HRV, and HR, regulates blood pressure. Notably, there is mounting evidence that the baroreflex plays a significant role in the regulation of blood pressure over a long period of time. An extensive range of mental and physical disorders and diseases, including internalizing, externalizing, and psychotic disorders as well as cardiometabolic diseases like diabetes, coronary heart disease, and hypertension, are linked to autonomic imbalance, which is characterized by high SNS and low PNS tones [31,32,33,34].

## 4. HRV Reactivity and Psychological Stressors

Sloan et al. examined the relationship between the RR interval, HRV responses, and recurring diary entries evaluating physical positions, negative emotions, and the time of day by analyzing 24 h electrocardiographic recordings from healthy subjects [39]. Their findings demonstrated, as predicted, that higher levels of stress were linked to lower RR intervals. Additionally, a significant rise in the LF/HF ratio was linked to psychological stress, indicating that there was higher SNS activity during stressful times of the day [39].

The frequency of occupational stresses, self-reported sleep quality, and daytime autonomic activity were examined by Kim et al. using a questionnaire survey and short-term HRV recordings of 223 healthy, male, white-collar workers. There was no discernible relationship between the five job stress levels and the HRV parameters [33]. Subsequent research, however, has revealed that a few HRV signs are indicative of psychological stress. The investigations varied in terms of the kind of stress-inducing task that was employed and the HRV reactivity (i.e., tasks conducted in a lab setting or subjective stress ratings). Furthermore, the investigations on HRV reactivity were either continuous 24 h studies or short-term (5 min) studies. Many studies found that HRV variables altered in response to the stress brought on by diverse techniques. Nonetheless, there was mixed evidence of considerable variation in the HRV measures. Low parasympathetic activity, which is defined by a drop in HF activity and an increase in LF activity, was the most reported component linked to variance in HRV variables. According to a recent study by Dimitriev et al., mental stress increases predictability, regularizes RR intervals, and decreases complexity [34]. This indicates a shift in HR behavior under stress toward periodicity and stability. Potential susceptibility to further stress is increased by decreased HRV and suppressed parasympathetic activation.

Emotional control affects how decisions are made. In a study with 63 young (mean age of 19) and 62 elderly (mean age of 72) volunteers, resting heart rate variability (HRV) and eye tracking were assessed while pictures of faces displaying a variety of emotions were shown [34]. Older participants with high HRV were more likely to avoid angry faces, suggesting a greater ability to minimize negative influences and a stronger positivity effect that can benefit emotional regulation and health. The younger participants, regardless of HRV level, showed no preference between happy or angry faces [40].

HRV measurement and functional MRI (fMRI) were employed in another investigation involving twenty young adults and twenty older persons. Their findings demonstrated that higher resting HRV (RMSSD) was linked with a better connection between the amygdala and the medial prefrontal cortex (mPFC) in all age groups, but only in the younger group between the amygdala and the ventrolateral PFC. Others authors show that, across age groups, increased HRV is linked with improved emotional regulation, with some variations in regional relationships over aging [41].

## 5. Brain–Heart Communication

Research has shown that HRV is the outcome of heart and brain communication, and that the ANS controls cardiac activity in a variety of situations to preserve the equilibrium between internal and external inputs. The neurological process responsible for generating HRV is the communication between the brain and the heart [30]. Brain–heart communication is referred to as an inherent cardiac nervous system. Its intricate structure is made up of proteins, ganglions, neurotransmitters, and supportive cells. Through the stimulation of the sympathetic and parasympathetic nervous systems, the nervous system sends signals that control heart processes.

Any stimuli (heart rate, rhythm, pressure, or hormonal variation) that alter the heart’s normal physiology cause the sensory neurons to send signals to the cardiac nervous system right away. This activates the sympathetic nervous system, which then sends an efferent nerve of the ANS through a descending pathway to regulate cardiac function appropriately [32]. In this cardiac brain communication, the heart’s intrinsic nervous system sends a signal to the vagus nerve’s ascending nerve fibers, the spinal column, the medulla, the hypothalamus, the thalamus, the amygdala, and finally, to the cerebral cortex for interpretation. The ganglion, which is in the peripheral region, receives signals from the brain’s myocardiocytes via the polysynaptic pathway. From there, the signals travel on to pre-ganglionic and premotor neurons. The activation of cardiac reflexes, such as baroreceptors, chemoreceptors, nasopharyngeal receptors, etc., as well as central autonomic reactions to stress, exercise, and sleep, regulate heart activities [33]. Clinical observations have shown that long-term neurodegenerative diseases may eventually impair ANS function and possibly cause it to fail. On the other hand, the pathological state of blood vessels, neurological issues, or lesions (inflammation or damage) may cause the ANS to become hyperactive [34].

## 6. Non-Invasive Brain Stimulation and HRV

Neuroimaging techniques have only lately been focused on comprehending these brain–body linkages, despite the brain’s importance in defining what is stressful for an individual [35]. The research to date indicates that the insular, anterior cingulate, and orbitofrontal brain areas function together as a network to interpret environmental inputs for motivational purposes and facilitate adaptive cardiovascular responses to stresses [36,37]. There is a larger corpus of the literature describing the mutual connections between the autonomic nervous system and certain brain regions, known as the Central Autonomic Network [28]. When considered collectively, these results demonstrated the central role of the brain in the pathogenesis of essential hypertension [38]. Indeed, alterations in brain function, structure, and organization are correlated with the existence of hypertension early in its course, and the brain is implicated in the onset of high blood pressure.

However, most studies conducted in this area are correlational, and the manipulation of experimental data is required to further our understanding of the causal relationship between cardiovascular responses and brain functioning. Non-invasive brain stimulation (NIBS) approaches present an intriguing means of exploring this kind of interaction. When it comes to the cardiovascular and autonomic systems, transcranial direct current stimulation (tDCS) and repeated transcranial magnetic stimulation (rTMS) have received the greatest research attention to date among the NIBS approaches. Both anodal and high-frequency rTMS could increase the excitability of the stimulated hemisphere, whereas low-frequency rTMS and cathodal tDCS could decrease it.

TMS is a non-invasive technique that depolarizes neurons and produces observable effects by electrically stimulating brain tissue in single pulses [30]. The excitability of the cerebral cortex can be altered by repetitive TMS using trains of impulses at the stimulated site and at distant locations along functional anatomical connections [42]. TDCS is distinct from TMS in that it modifies spontaneous neural network activity by applying mild electrical currents to various cortical regions, as opposed to inducing neuronal firing through supra-threshold neuronal membrane depolarization. The production of polarity-dependent alterations in cortical excitability is the main mechanism of action at the neuronal level [33]. Research on both humans and animals has shed light on the mechanisms behind tDCS’s effects on neuroplasticity and has demonstrated that, depending on the targeted brain areas, tDCS could cause alterations in motor activity, psychophysiological, and neuropsychological functioning [41]. Although there is a growing amount of data regarding NIBS’s effects on the cardiovascular system, these effects have not yet been methodically measured. The topic has only been the subject of one systematic review to date, which pointed out the heterogeneity of the included research and the lack of clear information about the effects of NIBS on the autonomic and cardiovascular systems [42]. This is mostly because most of the studies that are currently in existence were not intended to study brain–heart connections, but rather to comprehend the safety of NIBS using cardiovascular measures. Through a sequence of meta-analyses, we examined the correlation between NIBS and elevated HRV, BP, and HR. While HR and (marginally) BP were decreased, NIBS increased HRV. Moderator analysis indicated that even if impacts were modest, bigger effects might still be possible provided further research follows the precautions outlined below. Overall, the benefits were greater in trials that activated the PFC rather than the primary motor cortex or other brain regions (primarily Cz, T3, and pain-related areas) and utilized TMS rather than tDCS. First, with up-to-medium effects on HRV, TMS-based studies were more successful than tDCS-based ones. This outcome makes sense when we consider the technical challenges associated with tDCS in humans, since the electrical current produced into the cortex only goes a few centimeters deep in a non-selective manner. Because of its focality over cortical autonomic areas, TMS is theoretically a more suitable instrument to study ANS function [41].

Secondly, it seemed that stimulating the PFC was the ideal way to get impacts on the cardiovascular system. Regretfully, many of the reviewed trials, as indicated in Table 1, did not aim to alter cardiovascular function; instead, they only recorded heart rates and blood pressure for safety purposes. Due to the simplicity with which the motor-evoked potential may be obtained as an objective indicator of the efficacy of brain stimulation, most of the research on the effects of NIBS has generally concentrated on the motor cortex [39,43,44,45]. Given that this stimulation site was proven to be the least effective in reducing cardiovascular activation, this is an unpleasant situation. It should come as no surprise that the PFC was linked to the best stimulation in terms of HR decreases and HRV increases. The PFC is essential for the brain’s representation of both internal and external contexts, as well as the use of this knowledge to control behavior and peripheral physiology [46]. The PFC is implicated in the regulation of cardiovascular autonomic activity in both human and animal studies involving pharmacological blockades, lesions, and functional and structural neuroimaging. Furthermore, the PFC was found to be a central component in the regulation of HRV in a meta-analysis of brain areas linked to HRV [47]. The current results add to the body of work in this area and imply that it could be a useful area for future interventions related to physiological and psychological dysregulation.

## 7. Conclusions

This paper aimed to explore the intricate relationship between non-invasive brain stimulation (NIBS) techniques and the autonomic nervous system (ANS). Through a comprehensive review of the current literature, we examined the effects of various NIBS modalities, including transcranial magnetic stimulation (TMS) and transcranial electrical stimulation (tES), on autonomic function. Our findings suggest that NIBS holds the potential to modulate autonomic activity, influencing physiological processes such as heart rate variability (HRV), skin conductance, and sympathetic–parasympathetic balance. However, the exact mechanisms underlying these effects have yet to be fully elucidated. We discussed potential neural pathways and neurotransmitter systems involved in mediating the interaction between NIBS and the ANS. Furthermore, we highlighted clinical implications and potential therapeutic applications of harnessing NIBS-induced autonomic modulation, particularly in the context of neurological and psychiatric disorders characterized by autonomic dysfunction. Nonetheless, we emphasized the need for further research to better understand the nuances of this relationship, optimize stimulation parameters, and ensure safety and efficacy in clinical practice. Overall, this paper contributes to the growing body of knowledge regarding the complex interplay between brain stimulation techniques and autonomic regulation, paving the way for future investigations and clinical interventions aimed at harnessing the potential of NIBS for autonomic modulation. Out of the nearly 9000 NIBS (TMS/tDCS) articles that are now available in the literature, only 44 studies have examined the effects of NIBS on the ANS, and almost half of those studies have found a significant correlation between NIBS and modifications in autonomic tests. With the abundance of data on the cortical control of sympathetic and parasympathetic responses in animals, this was thought to be an unexpected discovery. For example, the lateral prefrontal cortex and the medial prefrontal cortex have distinct autonomic control. Moreover, rats’, cats’, and monkeys’ motor and premotor regions can all be stimulated electrically to produce distinct cardio-autonomic reactions. Finally, comparable cardiovascular responses are elicited by the stimulation of distinct insular cortex regions. In conclusion, this review showed that NIBS could induce different responses in the ANS. However, further studies are needed to confirm what has been seen in the literature and, above all, to identify standardized stimulation protocols. In conclusion, the effects of NIBS on the ANS, including its possible therapeutic benefits for autonomic diseases, have not yet been proven beyond a reasonable doubt. While both open-label and randomized controlled studies have frequently reported positive results, the unsystematic use of several treatment parameters, including location, frequency, intensity, and duration, has made it difficult to interpret the results and provided little guidance on which treatment parameters might be most helpful to modulate ANS parameters. To better understand the cortical control of ANS function and to identify new therapeutic approaches for ANS disorders, more studies focusing on ANS function as the primary outcome and NIBS on different brain regions with stimulation parameters are needed. Finally, cutting-edge stimulation techniques like theta burst TMS or primed 1 Hz stimulation may be advantageous for modifying ANS function since they may have longer-lasting impacts on neuroplasticity.

## Figures and Tables

**Table 1 biomedicines-12-00972-t001:** The time domain measures of heart rate variability.

Variable		Description
SSD	ms	Standard deviation of all NN intervals
SDANN	ms	Standard deviation of the averages of NN intervals in all 5 min segments of the entire recording
RMSSD	ms	The square root of the mean of the sum of the squares of differences between adjacent NN intervals
SDNN index	ms	Mean of the standard deviations of all NN intervals for all 5 min segments of the entire recording
SDSD	ms	Standard deviation of differences between adjacent NN intervals
pNN50	%	NN50 count divided by the total number of all NN intervals

**Table 2 biomedicines-12-00972-t002:** The frequency domain of heart rate variability (VLF: very low-frequency band, LF: low-frequency band, HF: high-frequency band).

Variable		Description
VLF	ms^2^	Power in a VLF range
LF	ms^2^	Power in a LF range
LF norm	ms^2^	LF power in normalized units of LF/(total power-VLF) × 100
HF	ms^2^	Power in a HF range
HF norm	nu	HF power in normalized units of HF/(total power-VLF) × 100
LF/HF		Ratio of LF (ms^2^)/HF (ms^2^)
Total Power		Variance of all NN intervals

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
