# Peer review of "Relationship between Non-Invasive Brain Stimulation and Autonomic Nervous System"

_biomedicines, 2024, doi:10.3390/biomedicines12050972_

Round 1
Reviewer 1 Report
Comments and Suggestions for Authors
Interesting and well written review. Here are my comments and suggestions for each paragraph.
Introduction
- Organization: The flow of ideas could be more structured. For example, the introduction starts with stress, then moves to the HPA axis and sympathetic nervous system, then to executive functions and the prefrontal cortex (PFC), and finally to heart rate variability (HRV) and non-invasive brain stimulation (NIBS). Reorganizing the paragraphs to have a more logical flow would improve readability.
- Clarity: Some sentences are long and complex, which can make it difficult for the reader to follow the main ideas. Breaking up long sentences and simplifying the language could enhance clarity.
- Transitions: The transitions between some paragraphs are abrupt. Adding transitional phrases or sentences could help the reader understand how the ideas are connected.
- Relevance: While the introduction covers a wide range of topics, some of the information may not be directly relevant to the main focus of the review, which seems to be the relationship between HRV and NIBS. Streamlining the introduction to focus on the most pertinent information would make it more effective.
- Conclusion: The introduction would benefit from a clear statement of the purpose and scope of the review at the end. This would help the reader understand what to expect in the following sections.
- Citations: The citation style "[1-3]" in lines 71 and 80 is inconsistent with the rest of the introduction. Ensuring consistency in citation style throughout the manuscript would improve its professional appearance.
Chapter 2: Stress and HRV
- This chapter provides a good overview of the concept of stress and the use of heart rate variability (HRV) as a measure of stress.
- Tables 1 and 2 are helpful in defining the different time-domain and frequency-domain measures of HRV.
- However, the chapter could benefit from a more in-depth explanation of how HRV specifically relates to stress. What are the physiological mechanisms linking stress to changes in HRV?
- Additionally, a brief discussion of the advantages and limitations of using HRV as a stress biomarker compared to other measures (e.g., cortisol) would strengthen this chapter.
Chapter 3: Psychological stress and autonomic regulation
- This chapter delves into the neural pathways involved in the stress response, focusing on the hypothalamic-pituitary-adrenal (HPA) axis and autonomic nervous system (ANS).
- The explanation of the interplay between the sympathetic and parasympathetic branches of the ANS in response to stress is informative.
- However, the chapter could be improved by providing more specific examples or studies illustrating how psychological stress alters autonomic regulation.
- The mention of the baroreceptor reflex is useful, but additional details on how this reflex is impacted by stress would enhance the reader's understanding.
Chapter 4: HRV reactivity and psychological stressors
- This chapter presents an overview of research examining HRV reactivity to various psychological stressors.
- The distinction between studies using lab-based stress tasks versus subjective stress ratings is important to note.
- However, the chapter would benefit from a more systematic review of the literature. What types of stressors have been studied? What are the consistent findings across studies? Are there any contradictory results that need to be addressed?
- Additionally, a clearer summary of the main conclusions that can be drawn from this body of research would improve the chapter. What does the evidence tell us about the utility of HRV as a marker of psychological stress reactivity?
Conclusion
- The conclusion could benefit from a more explicit statement of the key takeaways from the review. What are the most important findings or implications that the authors want readers to remember?
- While the conclusion mentions the need for further research, it could be more specific about the gaps in current knowledge and the priority areas for future investigation. What are the critical unanswered questions or challenges that need to be addressed?
- The conclusion could also discuss potential limitations of the reviewed studies and how these limitations might impact the interpretation or generalizability of the findings.
- The paragraph starting with "Out of the nearly 9000 NIBS (TMS/tDCS) articles..." seems to introduce new information that was not discussed in detail in the main body of the review. It might be better to integrate this information into the appropriate sections earlier in the manuscript or remove it from the conclusion.
- The final sentence, "However, further studies are needed to confirm what has been seen in the literature and above all to identify standardized stimulation protocols," is a bit vague. What specific aspects of the literature need to be confirmed? What considerations should be taken into account when developing standardized stimulation protocols?
Author Response
Reviewer 1
Interesting and well written review. Here are my comments and suggestions for each paragraph.
Reply
Dear reviewer, thank you for your comment, your suggestions will greatly improve our manuscript.
Reviewer 1
Introduction
- Organization: The flow of ideas could be more structured. For example, the introduction starts with stress, then moves to the HPA axis and sympathetic nervous system, then to executive functions and the prefrontal cortex (PFC), and finally to heart rate variability (HRV) and non-invasive brain stimulation (NIBS). Reorganizing the paragraphs to have a more logical flow would improve readability.
- Clarity: Some sentences are long and complex, which can make it difficult for the reader to follow the main ideas. Breaking up long sentences and simplifying the language could enhance clarity.
- Transitions: The transitions between some paragraphs are abrupt. Adding transitional phrases or sentences could help the reader understand how the ideas are connected.
- Relevance: While the introduction covers a wide range of topics, some of the information may not be directly relevant to the main focus of the review, which seems to be the relationship between HRV and NIBS. Streamlining the introduction to focus on the most pertinent information would make it more effective.
- Conclusion: The introduction would benefit from a clear statement of the purpose and scope of the review at the end. This would help the reader understand what to expect in the following sections.
- Citations: The citation style "[1-3]" in lines 71 and 80 is inconsistent with the rest of the introduction. Ensuring consistency in citation style throughout the manuscript would improve its professional appearance.
Reply
Dear reviewer, thank you for your suggestion. We have renstructured the introduction section of our manuscript. Below you will find the new paragarph added:
Noninvasive brain stimulation (NIBS) methods have grown in significance in cognitive neuroscience during the last thirty years. By establishing causal relationships between cortical brain regions and their corresponding cognitive, emotional, sensory, and motor processes, they made important advancements [1]. In this review we assess the effect of NIBS on various markers of autonomic nervous system activity, such as heart rate (HR) and various heart rate variability (HRV) parameters [2].
To design future research that aim to examine the fundamental processes of cortical modulation of autonomic nervous system (ANS) activities, it is imperative to comprehend whether and how NIBS should be employed to modulate HRV and HR. The interaction of multiple cortical regions, such as the insular cortex and medial prefrontal cortex (MPFC), higher subcortical regions, such as the amygdala and the bed nucleus of the stria terminalis (BNST), multiple areas and nuclei of the hypothalamus (e.g., paraventricular nucleus, dorsomedial hypothalamic nucleus), the periaqueductal gray (PAG), and brainstem regions, such as the parabrachial nucleus (PBN), the solitarius tract (NTS), the nucleus ambiguous (NA), the area postrema, the locus coeruleus, and the dorsal motor nucleus of the vagus (DMNV), and the rostral (RVLM) and caudal (CVLM) ventrolateral medulla. These interconnected regions come together to form the central autonomic nervous system (CAN) [3,4]. Thus, by integrating both cortical perceptual representations of one's bodily or visceral states and conceptual interpretations of this perceptual input, the MPFC, including the anterior cingulate cortex (ACC) and the insular cortex, are involved in the regulation of hierarchically lower level regions of the CAN such as the amygdala [5]. The entire cortex and subcortical regions share strong connections with the amygdala, which has been shown to be vitally engaged in identifying novel and important stimuli [6]. Efferent projections to the hypothalamic nuclei and the PAG initiate autonomic and behavioral responses to the perceived stimuli, while afferent outputs regulate attention and cognitive processing of those stimuli [7,8]. Bidirectional connections allow the hypothalamus and PAG to transmit signals from the amygdala to the medulla and lower brainstem nuclei [9]. Apart from projections originating from the PAG and hypothalamus, the brainstem also receives visceral afferents that travel via cranial, sacral, and thoracolumbar neural pathways to reach the NTS. The insula, the amygdala, and the hypothalamus receive the signals after they are sent to the PBN [1].
Some of the brain activity that was previously thought to be exclusively related to cognitive functions may be related to autonomic processing. Autonomic processing, via cortico-subcortical pathways, generates physiological responses for behavior that is contextually adaptive and suitable for these higher order functions [10,11]. An excellent method for explaining the connection between HRV and cognitive functions is the neurovisceral integration model, which postulates that inhibitory pathways in the pre-frontal cortex regulate subcortical cardioaccelerators circuits of the CAN in response to activity in various pre-frontal cortex regions during emotion, self-regulation and stress.
Prior research has demonstrated that, through network-level effects, the effects of NIBS extend beyond the cortical target regions to subcortical regions such the striatum or thalamus [12,13]. NIBS offers a promising chance to methodically investigate the underlying mechanisms of correlations between neuronal activity and both HR and HRV found in early brain imaging studies, since it can modulate the neuronal activity of both cortico-subcortical networks and circumscribed cortical regions [1].
Reviewer 1
Chapter 2: Stress and HRV
- This chapter provides a good overview of the concept of stress and the use of heart rate variability (HRV) as a measure of stress.
- Tables 1 and 2 are helpful in defining the different time-domain and frequency-domain measures of HRV.
- However, the chapter could benefit from a more in-depth explanation of how HRV specifically relates to stress. What are the physiological mechanisms linking stress to changes in HRV?
- Additionally, a brief discussion of the advantages and limitations of using HRV as a stress biomarker compared to other measures (e.g., cortisol) would strengthen this chapter.
Reply
Dear reviewer, thank you for your suggestion. We have renstructured the chapter 2. Below you will find the new paragarph added:
The hypothalamic-pituitary-adrenal (HPA) axis and the sympathetic nervous system (SNS) are the two primary mechanisms via which psychological stress affects the body [29]. There is strong coordination and connectivity between the ANS and HPA axis [30]. Via the sympathetic nervous system (SNS) and parasympathetic nervous system (PNS), the ANS rapidly induces physiological changes. By removing the inhibitory effect, the PNS encourages the sympathetic reaction to stress, also known as the "fight or flight" response [31]. The release of noradrenaline from the locus coeruleus is one of the subsequent alterations [32]. The HPA axis initiates a cascade of endocrine modifications during the stress response, commencing with the hypothalamus' production of corticotropin-releasing hormone [33]. By blocking the SNS and HPA axis, the PNS contributes significantly to reducing an individual's stress response [8]. Autonomic activity fluctuations linked to stress interfere with homeostatic functions [30]. Both external stimuli and the demands of the internal viscera are met by the ANS. While the stress response gives priority to external stimuli above internal needs, homeostasis is linked to the regulation of interior viscera. Therefore, stress happens when the PNS is unable to sufficiently meet an organism's physiological needs. As a result, parasympathetic tone measurement could be used as a stress and stress vulnerability index. Furthermore, stasis—a deficiency in endogenous variability in peripheral neurally controlled systems, such as heart rate—indicates a serious state of physiological distress [30].
The baroreceptor-heart rate reflex, a reciprocal alteration in ANS activity, serves as an example of the homeostatic functions of the ANS [34]. At the brainstem level, baroreceptor reflexes are primarily arranged within lower central autonomic substrates. However, these lower autonomic systems were merged with higher neural networks as a result of the rostral brain systems' evolutionary growth [32]. It has been demonstrated that the limbic system and forebrain, which include the hypothalamus, amygdala, and medial prefrontal cortex, send synaptic projections to both autonomic areas and brainstem reflex networks [31]. Baroreceptor reflex set points can be shifted or inhibited as a result of stressors [34]. The SNS and PNS are the two autonomic branches that are reciprocally controlled by brainstem baroreceptor reflexes [32]. Depending on the situation, the reciprocal method of autonomic control may change. Berntson et al. [33] produced different patterns of control with orthostatic challenges and normal psychological stressors using selective pharmacological blockades targeting the SNS and PNS.
Individual differences in responses to orthostatic stress were negligible. On the other hand, autonomic responsiveness varied widely among individuals in reaction to psychological stressors. Participants exhibited a reciprocal pattern of autonomic response, sympathetic activation, or vagal withdrawal in response to psychological stressors [33]. These results demonstrate that people react differently to psychological pressures.
Reviewer 1
Chapter 3: Psychological stress and autonomic regulation
- This chapter delves into the neural pathways involved in the stress response, focusing on the hypothalamic-pituitary-adrenal (HPA) axis and autonomic nervous system (ANS).
- The explanation of the interplay between the sympathetic and parasympathetic branches of the ANS in response to stress is informative.
- However, the chapter could be improved by providing more specific examples or studies illustrating how psychological stress alters autonomic regulation.
- The mention of the baroreceptor reflex is useful, but additional details on how this reflex is impacted by stress would enhance the reader's understanding.
Reply
Dear reviewer, thank you for your suggestion. We have renstructured the chapter 3. Below you will find the new paragarph added:
There is copious and comprehensive evidence supporting the involvement of the ANS in stress and health. The two main branches of the autonomic nervous system (ANS) are the parasympathetic nervous system (PNS), which is linked to rest and digest and vegetative and restorative processes, and the sympathetic nervous system (SNS), which is involved with energy mobilization and the fight-or-flight response. These systems are often in a state of dynamic equilibrium when health is good, with the PNS predominating. But as previously mentioned, an imbalance resulting in continuous activation of fight-or-flight reactions under stressful situations can cause excessive strain on physiological systems (allostatic load) [30]. The baroreflex is one mechanism that connects the ANS and blood preassure [32]. To maintain blood flow to essential organs like the brain and heart, pressure-sensitive receptors in the carotid and aortic arches sense increases and decreases in blood pressure. These receptors then transmit those signals to the brain, which causes reflex adjustments in blood pressure through the regulation of sympathetic and parasympathetic outflows. Thus, the baroreflex is how ANS activity, as measured by myocardial contractility, peripheral vascular resistance, HRV, and HR, regulates blood pressure. Notably, there is mounting evidence that the baroreflex plays a significant role in the regulation of blood pressure over the long period. An extensive range of mental and physical disorders and diseases, including internalizing, externalizing, and psychotic disorders as well as cardiometabolic diseases like diabetes, coronary heart disease, and hypertension, are linked to autonomic imbalance, which is characterized by high SNS and low PNS tones [31-34].
Reviewer 1
Chapter 4: HRV reactivity and psychological stressors
- This chapter presents an overview of research examining HRV reactivity to various psychological stressors.
- The distinction between studies using lab-based stress tasks versus subjective stress ratings is important to note.
- However, the chapter would benefit from a more systematic review of the literature. What types of stressors have been studied? What are the consistent findings across studies? Are there any contradictory results that need to be addressed?
- Additionally, a clearer summary of the main conclusions that can be drawn from this body of research would improve the chapter. What does the evidence tell us about the utility of HRV as a marker of psychological stress reactivity?
Reply
Dear reviewer, thank you for your suggestion. We have renstructured the chapter 4. Below you will find the new paragarph added:
Emotional control affects how decisions are made. In research with 63 youthful (mean age of 19) and 62 elderly (mean age of 72) volunteers, resting heart rate variability (HRV) and eye tracking were assessed while pictures of faces displaying a variety of emotions were shown [43]. Older participants with high HRV were more likely to avoid angry faces, suggesting a greater ability to minimize negative influences and a stronger positivity effect that can benefit emotional regulation and health. Young participants, regardless of HRV level, showed no preference between happy or angry faces [44].
HRV and functional MRI (fMRI) were employed in another investigation involving twenty young adults and twenty older persons. Their findings demonstrated that higher resting HRV (RMSSD) was linked to better connection between the amygdala and the medial prefrontal cortex (mPFC) in all age groups, but only in the younger group between the amygdala and the ventrolateral PFC. Others authors show that, across age groups, increased HRV is linked to improved emotional regulation, with some variations in regional relationships over aging [45].
Reviewer 1
Conclusion
- The conclusion could benefit from a more explicit statement of the key takeaways from the review. What are the most important findings or implications that the authors want readers to remember?
- While the conclusion mentions the need for further research, it could be more specific about the gaps in current knowledge and the priority areas for future investigation. What are the critical unanswered questions or challenges that need to be addressed?
- The conclusion could also discuss potential limitations of the reviewed studies and how these limitations might impact the interpretation or generalizability of the findings.
- The paragraph starting with "Out of the nearly 9000 NIBS (TMS/tDCS) articles..." seems to introduce new information that was not discussed in detail in the main body of the review. It might be better to integrate this information into the appropriate sections earlier in the manuscript or remove it from the conclusion.
- The final sentence, "However, further studies are needed to confirm what has been seen in the literature and above all to identify standardized stimulation protocols," is a bit vague. What specific aspects of the literature need to be confirmed? What considerations should be taken into account when developing standardized stimulation protocols?
Reply
Dear reviewer, thank you for your suggestion. We have implemented the conclusion section. Below you will find the new paragarph added:
In conclusion, the effects of NIBS on ANS, including its possible therapeutic benefit for autonomic diseases, have not yet been proven beyond a reasonable doubt. While both open-label and randomized controlled studies have frequently reported positive results, the unsystematic use of several treatment parameters, including location, frequency, intensity, and duration, has made it difficult to interpret the results and provided little guidance on which treatment parameters might be most helpful to modulate ANS parameters. To better understand cortical control of ANS function and to identify new therapeutic approaches for ANS disorders, more research focusing on ANS function as the primary outcome and NIBS on different brain regions with stimulation parameters are needed. Finally, cutting-edge stimulation techniques like theta burst TMS stimulation or primed 1 Hz stimulation may be advantageous for modifying ANS function since they may have longer-lasting impacts on neuroplasticity.
Reviewer 2 Report
Comments and Suggestions for Authors
In this paper, the authors summarized the state of knowledge and some research done on the relationship between non-invasive brain stimulation (NIBS) and he autonomic nervous system (ANS). The composition of the paper included an Introduction Section that discussed stress and the stress response, Section 2 dealt with stress and heart rate variability (HRV), Section 3 covered psychological stress and autonomic regulation, Section 4 was HRV and psychological stressors, and Section 5 covered brain-heart communication. Finally, in section 6 NIBs (rTMS and tDCS) and HRV were addressed followed by overall conclusions.
Overall, the paper provided some good background information on the above topics and some of the relationships between them. The paper was also well-written with only a few formatting mistakes or instances of awkward wording. I do have some concerns about the overall composition and goals of the paper along with questions about what type of paper this was supposed to be. These concerns are below.
1. I am having trouble figuring out what type of article this was and what the authors were trying to accomplish. First, at the top of the first page for “Type of the Paper” it says that it is an “Article”. However, it is not an original research article. Should “Review” have been written there???
Later on in the paper, the authors state that it was a review article. For instance, line 120. If it was a review it did review some studies but not in detail. Overall, it read more like a textbook than a review article. Single studies or sets of studies on a topic were never explained in depth. There was no table of studies to see or any type of data figure at all. Overall, it seemed like a textbook and not a review.
Deeper in the paper in line 290 the authors state the performed a sequence of meta-analyses. Are they talking about in the past? In this paper, there are almost none of the features of a meta-analysis.
Overall, this paper seems more like a viewpoint paper than anything or as mentioned before a textbook writing style. The papers that are discussed are not done so in detail, only basic overall concepts and physiology are discussed. There are no tables of all the studies done on a topic or any figures of key overall findings from groups of studies or representative studies.
Thus, it is hard to evaluate this paper based on the above issues.
2. Although the paper is well-written overall there are some minor formatting mistakes a few instances of awkward wording. There are too many to list individually. A few are
-line 23 “greatest” should read “most” or something similar.
-line 86 too many spaces after autonomic
-line 120 to 125 paragraph has different between line spacing that the other paragraphs
-line 137 “Numerous research” should read “Numerous research studies” or something similar.
-Line 183 tab the paragraph?
-Line 228 needs to be reworded, hard to understand what is meant
-Line 267 it is repetitive TMS not repeated
-Bibliography has many errors. Some journal titles are abbreviated such as Ref 1 and 15 and others. Others are abbreviated with periods after each word such as Refs 3 and 4 and others, some are spelled all the way out such as Ref 16, 17, 19, and others. Etc etc
Comments on the Quality of English Languagemoderate spell check, wording, and formatting
Author Response
Reviewer 2
In this paper, the authors summarized the state of knowledge and some research done on the relationship between non-invasive brain stimulation (NIBS) and he autonomic nervous system (ANS). The composition of the paper included an Introduction Section that discussed stress and the stress response, Section 2 dealt with stress and heart rate variability (HRV), Section 3 covered psychological stress and autonomic regulation, Section 4 was HRV and psychological stressors, and Section 5 covered brain-heart communication. Finally, in section 6 NIBs (rTMS and tDCS) and HRV were addressed followed by overall conclusions.
Overall, the paper provided some good background information on the above topics and some of the relationships between them. The paper was also well-written with only a few formatting mistakes or instances of awkward wording. I do have some concerns about the overall composition and goals of the paper along with questions about what type of paper this was supposed to be. These concerns are below.
Reply
Dear reviewer, thank you for your comment, your suggestions will greatly improve our manuscript.
Reviewer 2
- I am having trouble figuring out what type of article this was and what the authors were trying to accomplish. First, at the top of the first page for “Type of the Paper” it says that it is an “Article”. However, it is not an original research article. Should “Review” have been written there???
Later on in the paper, the authors state that it was a review article. For instance, line 120. If it was a review it did review some studies but not in detail. Overall, it read more like a textbook than a review article. Single studies or sets of studies on a topic were never explained in depth. There was no table of studies to see or any type of data figure at all. Overall, it seemed like a textbook and not a review.
Deeper in the paper in line 290 the authors state the performed a sequence of meta-analyses. Are they talking about in the past? In this paper, there are almost none of the features of a meta-analysis.
Overall, this paper seems more like a viewpoint paper than anything or as mentioned before a textbook writing style. The papers that are discussed are not done so in detail, only basic overall concepts and physiology are discussed. There are no tables of all the studies done on a topic or any figures of key overall findings from groups of studies or representative studies.
Thus, it is hard to evaluate this paper based on the above issues.
Reply
Dear reviewer, thank you for your valuable comments. We have completely remodeled our manuscript following your suggestions and, therefore, we have restructured the introduction, paragraphs 2 to 4, and the conclusions. I also specify that our manuscript is a narrative review. We apologize for the typo at the beginning of the manuscript, we have changed the typology from "Article" to "review". We have highlighted all the parts included in the manuscript.
Below are all the changes we made:
Introduction
Noninvasive brain stimulation (NIBS) methods have grown in significance in cognitive neuroscience during the last thirty years. By establishing causal relationships between cortical brain regions and their corresponding cognitive, emotional, sensory, and motor processes, they made important advancements [1]. In this review we assess the effect of NIBS on various markers of autonomic nervous system activity, such as heart rate (HR) and various heart rate variability (HRV) parameters [2].
To design future research that aim to examine the fundamental processes of cortical modulation of autonomic nervous system (ANS) activities, it is imperative to comprehend whether and how NIBS should be employed to modulate HRV and HR. The interaction of multiple cortical regions, such as the insular cortex and medial prefrontal cortex (MPFC), higher subcortical regions, such as the amygdala and the bed nucleus of the stria terminalis (BNST), multiple areas and nuclei of the hypothalamus (e.g., paraventricular nucleus, dorsomedial hypothalamic nucleus), the periaqueductal gray (PAG), and brainstem regions, such as the parabrachial nucleus (PBN), the solitarius tract (NTS), the nucleus ambiguous (NA), the area postrema, the locus coeruleus, and the dorsal motor nucleus of the vagus (DMNV), and the rostral (RVLM) and caudal (CVLM) ventrolateral medulla. These interconnected regions come together to form the central autonomic nervous system (CAN) [3,4]. Thus, by integrating both cortical perceptual representations of one's bodily or visceral states and conceptual interpretations of this perceptual input, the MPFC, including the anterior cingulate cortex (ACC) and the insular cortex, are involved in the regulation of hierarchically lower level regions of the CAN such as the amygdala [5]. The entire cortex and subcortical regions share strong connections with the amygdala, which has been shown to be vitally engaged in identifying novel and important stimuli [6]. Efferent projections to the hypothalamic nuclei and the PAG initiate autonomic and behavioral responses to the perceived stimuli, while afferent outputs regulate attention and cognitive processing of those stimuli [7,8]. Bidirectional connections allow the hypothalamus and PAG to transmit signals from the amygdala to the medulla and lower brainstem nuclei [9]. Apart from projections originating from the PAG and hypothalamus, the brainstem also receives visceral afferents that travel via cranial, sacral, and thoracolumbar neural pathways to reach the NTS. The insula, the amygdala, and the hypothalamus receive the signals after they are sent to the PBN [1].
Some of the brain activity that was previously thought to be exclusively related to cognitive functions may be related to autonomic processing. Autonomic processing, via cortico-subcortical pathways, generates physiological responses for behavior that is contextually adaptive and suitable for these higher order functions [10,11]. An excellent method for explaining the connection between HRV and cognitive functions is the neurovisceral integration model, which postulates that inhibitory pathways in the pre-frontal cortex regulate subcortical cardioaccelerators circuits of the CAN in response to activity in various pre-frontal cortex regions during emotion, self-regulation and stress.
Prior research has demonstrated that, through network-level effects, the effects of NIBS extend beyond the cortical target regions to subcortical regions such the striatum or thalamus [12,13]. NIBS offers a promising chance to methodically investigate the underlying mechanisms of correlations between neuronal activity and both HR and HRV found in early brain imaging studies, since it can modulate the neuronal activity of both cortico-subcortical networks and circumscribed cortical regions [1].
Paragraph 2
The hypothalamic-pituitary-adrenal (HPA) axis and the sympathetic nervous system (SNS) are the two primary mechanisms via which psychological stress affects the body [29]. There is strong coordination and connectivity between the ANS and HPA axis [30]. Via the sympathetic nervous system (SNS) and parasympathetic nervous system (PNS), the ANS rapidly induces physiological changes. By removing the inhibitory effect, the PNS encourages the sympathetic reaction to stress, also known as the "fight or flight" response [31]. The release of noradrenaline from the locus coeruleus is one of the subsequent alterations [32]. The HPA axis initiates a cascade of endocrine modifications during the stress response, commencing with the hypothalamus' production of corticotropin-releasing hormone [33]. By blocking the SNS and HPA axis, the PNS contributes significantly to reducing an individual's stress response [8]. Autonomic activity fluctuations linked to stress interfere with homeostatic functions [30]. Both external stimuli and the demands of the internal viscera are met by the ANS. While the stress response gives priority to external stimuli above internal needs, homeostasis is linked to the regulation of interior viscera. Therefore, stress happens when the PNS is unable to sufficiently meet an organism's physiological needs. As a result, parasympathetic tone measurement could be used as a stress and stress vulnerability index. Furthermore, stasis—a deficiency in endogenous variability in peripheral neurally controlled systems, such as heart rate—indicates a serious state of physiological distress [30].
The baroreceptor-heart rate reflex, a reciprocal alteration in ANS activity, serves as an example of the homeostatic functions of the ANS [34]. At the brainstem level, baroreceptor reflexes are primarily arranged within lower central autonomic substrates. However, these lower autonomic systems were merged with higher neural networks as a result of the rostral brain systems' evolutionary growth [32]. It has been demonstrated that the limbic system and forebrain, which include the hypothalamus, amygdala, and medial prefrontal cortex, send synaptic projections to both autonomic areas and brainstem reflex networks [31]. Baroreceptor reflex set points can be shifted or inhibited as a result of stressors [34]. The SNS and PNS are the two autonomic branches that are reciprocally controlled by brainstem baroreceptor reflexes [32]. Depending on the situation, the reciprocal method of autonomic control may change. Berntson et al. [33] produced different patterns of control with orthostatic challenges and normal psychological stressors using selective pharmacological blockades targeting the SNS and PNS.
Individual differences in responses to orthostatic stress were negligible. On the other hand, autonomic responsiveness varied widely among individuals in reaction to psychological stressors. Participants exhibited a reciprocal pattern of autonomic response, sympathetic activation, or vagal withdrawal in response to psychological stressors [33]. These results demonstrate that people react differently to psychological pressures.
Paragraph 3
There is copious and comprehensive evidence supporting the involvement of the ANS in stress and health. The two main branches of the autonomic nervous system (ANS) are the parasympathetic nervous system (PNS), which is linked to rest and digest and vegetative and restorative processes, and the sympathetic nervous system (SNS), which is involved with energy mobilization and the fight-or-flight response. These systems are often in a state of dynamic equilibrium when health is good, with the PNS predominating. But as previously mentioned, an imbalance resulting in continuous activation of fight-or-flight reactions under stressful situations can cause excessive strain on physiological systems (allostatic load) [30]. The baroreflex is one mechanism that connects the ANS and blood preassure [32]. To maintain blood flow to essential organs like the brain and heart, pressure-sensitive receptors in the carotid and aortic arches sense increases and decreases in blood pressure. These receptors then transmit those signals to the brain, which causes reflex adjustments in blood pressure through the regulation of sympathetic and parasympathetic outflows. Thus, the baroreflex is how ANS activity, as measured by myocardial contractility, peripheral vascular resistance, HRV, and HR, regulates blood pressure. Notably, there is mounting evidence that the baroreflex plays a significant role in the regulation of blood pressure over the long te. An extensive range of mental and physical disorders and diseases, including internalizing, externalizing, and psychotic disorders as well as cardiometabolic diseases like diabetes, coronary heart disease, and hypertension, are linked to autonomic imbalance, which is characterized by high SNS and low PNS tones [31-34].
Paragraph 4
Emotional control affects how decisions are made. In research with 63 youthful (mean age of 19) and 62 elderly (mean age of 72) volunteers, resting heart rate variability (HRV) and eye tracking were assessed while pictures of faces displaying a variety of emotions were shown [43]. Older participants with high HRV were more likely to avoid angry faces, suggesting a greater ability to minimize negative influences and a stronger positivity effect that can benefit emotional regulation and health. Young participants, regardless of HRV level, showed no preference between happy or angry faces [44].
HRV and functional MRI (fMRI) were employed in another investigation involving twenty young adults and twenty older persons. Their findings demonstrated that higher resting HRV (RMSSD) was linked to better connection between the amygdala and the medial prefrontal cortex (mPFC) in all age groups, but only in the younger group between the amygdala and the ventrolateral PFC. Others aothors shows that, across age groups, increased HRV is linked to improved emotional regulation, with some variations in regional relationships over aging [45].
Conclusion
In conclusion, the effects of NIBS on ANS, including its possible therapeutic benefit for autonomic diseases, have not yet been proven beyond a reasonable doubt. While both open-label and randomized controlled studies have frequently reported positive results, the unsystematic use of several treatment parameters, including location, frequency, intensity, and duration, has made it difficult to interpret the results and provided little guidance on which treatment parameters might be most helpful to modulate ANS parameters. To better understand cortical control of ANS function and to identify new therapeutic approaches for ANS disorders, more research focusing on ANS function as the primary outcome and NIBS on different brain regions with stimulation parameters are needed. Finally, cutting-edge stimulation techniques like theta burst TMS stimulation or primed 1 Hz stimulation may be advantageous for modifying ANS function since they may have longer-lasting impacts on neuroplasticity.
- Although the paper is well-written overall there are some minor formatting mistakes a few instances of awkward wording. There are too many to list individually. A few are
-line 23 “greatest” should read “most” or something similar.
-line 86 too many spaces after autonomic
-line 120 to 125 paragraph has different between line spacing that the other paragraphs
-line 137 “Numerous research” should read “Numerous research studies” or something similar.
-Line 183 tab the paragraph?
-Line 228 needs to be reworded, hard to understand what is meant
-Line 267 it is repetitive TMS not repeated
-Bibliography has many errors. Some journal titles are abbreviated such as Ref 1 and 15 and others. Others are abbreviated with periods after each word such as Refs 3 and 4 and others, some are spelled all the way out such as Ref 16, 17, 19, and others. Etc etc
Reply
Dear reviewer, thank you for your valuable comments. We have corrected all errors in the text as you suggested.
Round 2
Reviewer 1 Report
Comments and Suggestions for Authors
The authors responded to my comments very well. Thank you.
Reviewer 2 Report
Comments and Suggestions for Authors
The authors have addressed my previous comments, fixed many typographical and formatting errors, and greatly improved the wording of the paper. So it is much improved. I would suggest the authors consider a table or two of the studies covered or a figure or two of some of the key findings of the most important studies on the topic but I guess this is not mandatory.
Comments on the Quality of English LanguageMuch improved.